# An Improved Gradient Boosting Regression Tree Estimation Model for Soil Heavy Metal (Arsenic) Pollution Monitoring Using Hyperspectral Remote Sensing

**Lifei Wei [1], Ziran Yuan [1,\*], Yanfei Zhong [2,\*] , Lanfang Yang [1], Xin Hu [2] and Yangxi Zhang [1]**

[1] Faculty of Resources and Environmental Science, Hubei University, Wuhan 430062, China; weilifei11@163.com (L.W.); lfyang@hubu.edu.cn (L.Y.); yangxi.zhang@hotmail.com (Y.Z.)

[2] State Key Laboratory of Information Engineering in Surveying, Mapping and Remote sensing, Wuhan University, Wuhan 430072, China; whu_huxin@whu.edu.cn

\* Correspondence: yuanziran11@163.com (Z.Y.); zhongyanfei@whu.edu.cn (Y.Z.)

**Abstract:** Hyperspectral remote sensing can be used to effectively identify contaminated elements in soil. However, in the field of monitoring soil heavy metal pollution, hyperspectral remote sensing has the characteristics of high dimensionality and high redundancy, which seriously affect the accuracy and stability of hyperspectral inversion models. To resolve the problem, a gradient boosting regression tree (GBRT) hyperspectral inversion algorithm for heavy metal (Arsenic (As)) content in soils based on Spearman's rank correlation analysis (SCA) coupled with competitive adaptive reweighted sampling (CARS) is proposed in this paper. Firstly, the CARS algorithm is used to roughly select the original spectral data. Second derivative (SD), Gaussian filtering (GF), and min-max normalization (MMN) pretreatments are then used to improve the correlation between the spectra and As in the characteristic band enhancement stage. Finally, the low-correlation bands are removed using the SCA method, and a subset with absolute correlation values greater than 0.6 is retained as the optimal band subset after each pretreatment. For the modeling, the five most representative characteristic bands were selected in the Honghu area of China, and the nine most representative characteristic bands were selected in the Daye area of China. In order to verify the generalization ability of the proposed algorithm, 92 soil samples from the Honghu and Daye areas were selected as the research objects. With the use of support vector machine regression (SVMR), linear regression (LR), and random forest (RF) regression methods as comparative methods, all the models obtained a good prediction accuracy. However, among the different combinations, CARS-SCA-GBRT obtained the highest precision, which indicates that the proposed algorithm can select fewer characteristic bands to achieve a better inversion effect, and can thus provide accurate data support for the treatment and recovery of heavy metal pollution in soils.

**Keywords:** soil heavy metal pollution; competitive adaptive reweighted sampling; gradient boosting regression tree; characteristic bands

## 1. Introduction

Heavy metal pollution in soil is caused by human activity that brings those metals into the soil, resulting in degradation of its quality and deterioration of the ecological environment. Heavy metals in soil are difficult to degrade, easy to accumulate, and toxic. Heavy metals in soil can be transferred to all the parts of plants through plant growth and absorption [1,2]. In soil they can also accumulate through the food chain and eventually enter the human body. Once in the bloodstream, heavy metals

can dissolve red blood cells, destroy normal cells, and are both carcinogenic and teratogenic. Arsenic (As) pollution can directly lead to a decrease in soil fertility and crop yield reduction. Furthermore, planting crops in contaminated soil or using contaminated surface water to irrigate crops causes the crops to absorb large amounts of toxic substances [3,4].

The traditional soil heavy metal detection methods are photometry, chemical analysis, atomic fluorescence spectrometry, inductively coupled plasma optical emission spectroscopy, and surface-enhanced Raman spectroscopy [5]. Although these methods can achieve a high level of precision, they do require a lot of manpower, material resources, equipment, and time. The development of remote sensing technology, especially the development of hyperspectral analysis technology, has made it possible to use continuous, high-resolution spectral bands to predict the heavy metal content in soil, and large-area rapid detection can be achieved, avoiding complex chemical analysis steps. Singh et al. [6] analyzed the iron (Fe), As, and copper (Cu) content of soil samples taken near Baoshan Mine in Hunan province, China. After measuring the spectral reflectance of the soil in laboratories and establishing a partial least squares (PLS)-based soil heavy metal concentration regression model, it was concluded that the soil heavy metal content could be indirectly evaluated by the soil spectral reflectance. Gholizadeh et al. [7] proposed a multivariate calibration method based on support vector machine regression (SVMR) to establish the relationship between visible-near-infrared (VIS-NIR) regional reflectance spectra and the manganese (Mn), Cu, cadmium (Cd), zinc (Zn), and lead (Pb) content in soil. Tian et al. [8] used soil samples to analyze the relationship between magnesium (Mg) content and soil reflectance spectra in soil samples, using principal component regression analysis, PLS regression analysis, and SVMR analysis. The modeling method was used to establish an estimation model of Mg content, and the final results showed that the estimation accuracy of the SVMR model was the highest. Ma et al. [9] studied the inversion of heavy metals in the soil of reclaimed mining areas, and they introduced extreme learning machine (ELM) technology to inversion modeling, and compared this with the traditional PLS regression and SVMR methods. The results of Ma et al. [6] showed that the ELM model can achieve a higher accuracy than the models of SVMR and PLS in the prediction of Zn, Cu, Cd, and Chromium (Cr).

Hyperspectral remote sensing monitoring in the field of soil heavy metal pollution has the characteristics of high dimensionality and high redundancy, which seriously affect the accuracy and stability of the hyperspectral inversion models. Characteristic band selection algorithms can be used in hyperspectral inversion models to improve the model predictive ability. Sun et al. [10] took the prediction of nickel (Ni) content in soils from Hengyang and Zhangzhou of Hunan province, China, as an example. The spectra of the heavy metal adsorption and immobilization materials in the soil were used to predict the heavy metal content with the genetic partial least squares method (GA-PLSR). This study laid a foundation for predicting soil heavy metal content using field-measured spectra and hyperspectral remote sensing images. Tan et al. [11] selected two research areas in metal mining and coal mining areas in northern China, and used competitive adaptive reweighted sampling-partial least squares (CARS-PLS) to select the characteristic bands. The results showed that, compared with other linear models, the nonlinear CARS-PLS-SVMR model obtained the highest accuracy in soil heavy metal estimation in the different mining areas by using appropriate spectral feature extraction in the preprocessed spectra. Zhang et al. [12] took the city of Xinzheng in Henan province, China, as the research object, and selected five common hyperspectral characteristic bands of heavy metals which passed the $P = 0.01$ significance test as the independent variables of the inversion model. They then adopted the fixed-effect variable coefficient model based on the ordinary least squares estimation (OLS) method, which indicated that this panel data model could be used for rapid hyperspectral monitoring of heavy metals in the cultivated soil of Xinzheng.

This paper takes the soils of the areas of the cities of Honghu and Daye in Hubei province, China, as the research objects, and describes the field soil sampling, laboratory chemical analysis, and spectral collection and pretreatment that was undertaken in this study. A characteristic band selection algorithm based on Spearman's rank correlation analysis (SCA) coupled with competitive adaptive

reweighted sampling (CARS) was used. In order to verify the generalization ability of the algorithm, gradient boosting regression tree (GBRT), random forest (RF), linear regression (LR), and support vector machine (SVM) models were used to build the model, and we compared the accuracies to find the best method for the inversion of heavy metals in the soil. This will be of great significance to the rapid monitoring of soil heavy metal content in this region, and will provide a new approach for large-scale real-time monitoring and early warning of soil heavy metal pollution.

## 2. Materials and Methods

### 2.1. Study Areas

We chose the cities of Daye and Honghu as the study areas, which are two typical areas of Jianghan Plain in Hubei province, China. The city of Daye (114°31′~115°20′E, 29°40′~30°15′N) is located in the southeast of Hubei province, on the south bank of the middle reaches of the Yangtze River. The area features a typical subtropical continental monsoon climate, with four distinct seasons, and sufficient water and heat for crop production. The annual average temperature is 16.9 °C, and the average annual precipitation is 1385.8 mm. The city area is mainly hilly, with an altitude of 120–200 m. The zonal soils in this area are mainly red soil and yellow-brown soil. The area is a production base of grain, cotton, and oil-bearing crops. The Daye area is rich in mineral resources, and features a number of coppers, iron, coal, and limestone mines [13]. At the same time, the mining has greatly damaged the ecological environment, and the farmland soil near the mining area has been seriously polluted. Long-term mining and metallurgy activities have caused the soil to accumulate different levels of heavy metals.

The city of Honghu (113°07′~114°05′E, 29°39′~30°12′N) is a county-level city in the municipal region of Jingzhou, Hubei province, in the middle and lower reaches of the Yangtze River, in the southeast of Jianghan Plain. The landform type in this area is mainly alluvial plain. The city has good soil quality, and the soil acidity-alkalinity is moderate, making the area suitable for planting crops. However, excessive application of pesticides and fertilizers has caused serious pollution in the area. The water quality of Honghu Lake has been severely degraded, and much of the cultivated land has been exposed to heavy metals such as Cr, Pb, and Cd. The two study areas are shown in Figure 1.

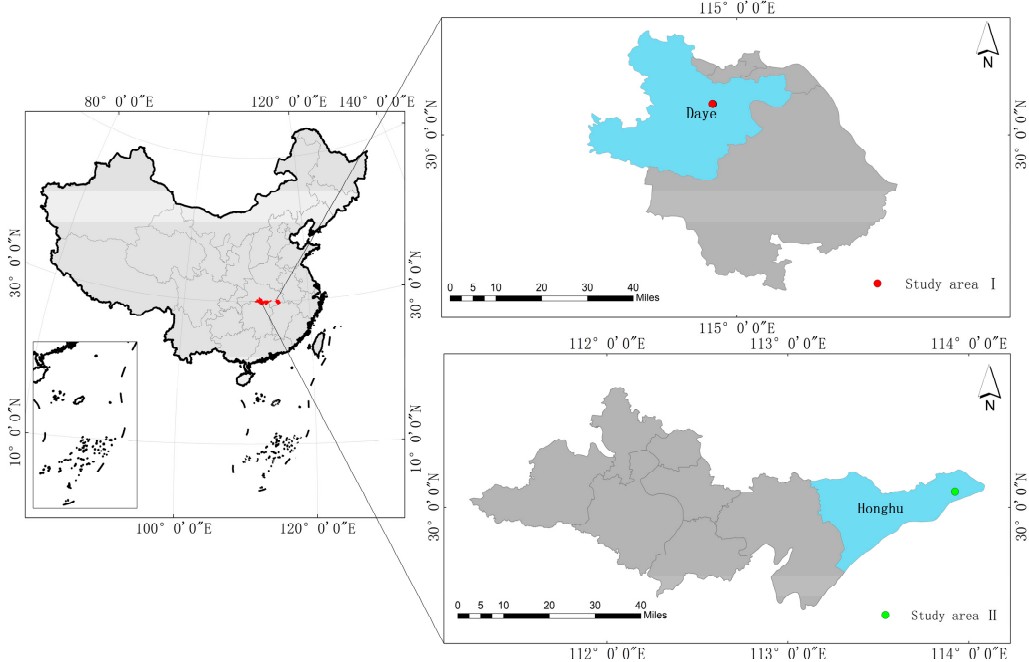

**Figure 1.** Study area locations. Study area I is Daye. Study area II is Honghu.

## 2.2. Research Methods

### 2.2.1. The CARS-SCA Characteristic Band Selection Algorithm

CARS is a new algorithm for spectral characteristic screening. Adaptive reweighted sampling (ARS) is used as the fitness function by the PLS method [14]. Cross-validation is used to optimize the calculation and select the optimal subset, i.e., the subset of the regression model with the highest precision. In addition, the variables with large error are eliminated, after multiple cycles of sampling to select the characteristic bands. There is a correlation between the characteristic bands of the spectra and the As content of the soil, which can be measured by the correlation coefficients. In this study, SCA was used as a correlation analysis tool between the spectra and the soil samples. Commonly used methods for amplifying characteristic bands are SD, Gaussian filtering (GF), and min-max normalization (MMN) pretreatments. In this study, a new method was used to combine the pretreatment methods of the spectra, to select the most suitable spectral bands. The specific process is as follows:

Step 1: The Monte Carlo sampling (MCS) method was used to randomly extract 80% of the sample set as a calibration set, and to establish a PLS regression model. $M_{l \times p}$ represents the $l$ sample and $p$ variable spectral matrix of the calibration set. $y_{l \times 1}$ represents the As content of the soil of the calibration set. $T$ represents the scoring matrix of $M$, and $B$ represents the $M$ and $T$ contact coefficient matrix. $c$ represents the regression coefficient vector of the PLS correction model established by $y$ and $T$. $f$ represents the prediction residual. The following relationships are established:

$$T = MB \tag{1}$$

$$y = Tc + f = MBc + f = Ah + f \tag{2}$$

where $h = Bc = [h_1, h_2, ..., h_p]^T$ is the p-dimensional coefficient vector. The absolute value of the ith element in $B$, denoted as $|h_i|(1 \leq i \leq p)$, reflects the ith wavelength's contribution to $y$. Therefore, the larger the value of $|h_i|$, the more important this variable is. The weight value is defined as:

$$w_i = \frac{h_i}{\sum_{i=1}^{p} |h_i|}, i = 1, 2, ..., p \tag{3}$$

We assume that there are P wavelength points in the entire spectral range, and that there are N MCSs in the CARS. As mentioned before, wavelength selection in CARS is a two-step process. In the first step, an exponentially decreasing function (EDF) is used to force the removal of wavelengths with relatively small absolute regression coefficients, At the ith MCS, the retention rate of the wavelength point can be obtained based on the following EDF:

$$r_i = ae^{-ki} \tag{4}$$

where $a$ and $k$ are two constants:

$$a = (p/2)^{1/(N-1)}, \ k = \frac{\ln(p/2)}{N-1} \tag{5}$$

N times of MCS can be used to obtain N PLS models. Each model is used to calculate one root-mean-square error of cross-validation (RMSECV) value. We finally select the subset of variables with the smallest RMSECV value, which is the optimal variable subset.

Step 2: The spectral set matrix $M$ is preprocessed. $M = [X_1, X_2, ..., X_i]$, $M$ is an i-dimensional vector of $X$, and $X_i$ is the spectral curve of each soil sample. $y_i$ represents the As content of the soil of the calibration set. According to the Spearman's rank correlation coefficient [15], the formula is as shown in Equation (6), and the calculation result is obtained as the correlation coefficient row vector. The corresponding $\rho$ is the Spearman's rank correlation coefficient of the corresponding bands, where the bands with higher correlation coefficients are selected.

$$\rho = \frac{\sum_{i}^{N}(x_i - \overline{x})(y_i - \overline{y})}{\left[\sum_{i-1}^{N}(x_i - \overline{x})^2 \sum_{i-1}^{N}(y_i - \overline{y})^2\right]^{\frac{1}{2}}} = 1 - \frac{6\sum d_i{}^2}{N(N^2 - 1)} \tag{6}$$

where N is the number of samples, and $d_i = x_i - y_i$ is the gradation difference of two paired variables.

Step 3: After GF, SD, and gaussian filtering again pretreatments, the processed data and the soil samples are separately subject to SCA to obtain the correlation coefficient row vectors:

(1) Gaussian filtering is a kind of linear smoothing filter which chooses weights according to the shape of the Gaussian function. It is very effective for suppressing noise obeying a normal distribution:

$$g(\chi) = \frac{1}{\sqrt{2\pi}\sigma} \exp\left[-\left(\frac{\chi}{2\sigma}\right)^2\right] \tag{7}$$

where $\chi$ is the distance of the weight function from the maximum point, and the scale parameter $\sigma$ represents the width of the Gaussian function, which determines the smoothness of the filtering.

(2) Second derivative pretreatment can eliminate some baseline and other background interference, while improving the spectral resolution and sensitivity. It is widely used in spectral analysis.

$$S(\lambda_i) = \frac{[\lambda_{i+1} - 2\lambda_i + \lambda_{i-1}]}{\Delta\lambda^2} \tag{8}$$

where $\lambda_i$ represents the reflectance value of the ith band, and $\Delta\lambda$ is the wavelength interval.

Step 4: After each treatment in step 3, SCA between the processed spectral values and soil heavy metal As is performed. The results of each analysis are then combined to obtain a correlation coefficient matrix $C(i \times 1)$. The bands with $|C_i| \geqq 0.6$ are then extracted. The data stored in the matrix $F(i \times q)$ corresponding to the GF, SD, and GF processing are then found, and the matrix $F(i \times q)$ is normalized to [0,1] using MMN, and the processed result is taken as the characteristic band subset.

(1) MMN can reduce the difference between the magnitudes and smooth the wave function of the data sample values.

$$x' = \frac{x_i - x_{\min}}{x_{\max} - x_{\min}} \tag{9}$$

where $x_{\max}$ is the maximum of $F_i$, and $x_{\min}$ is the minimum of $F_i$. The entire algorithm flow is shown in Figure 2.

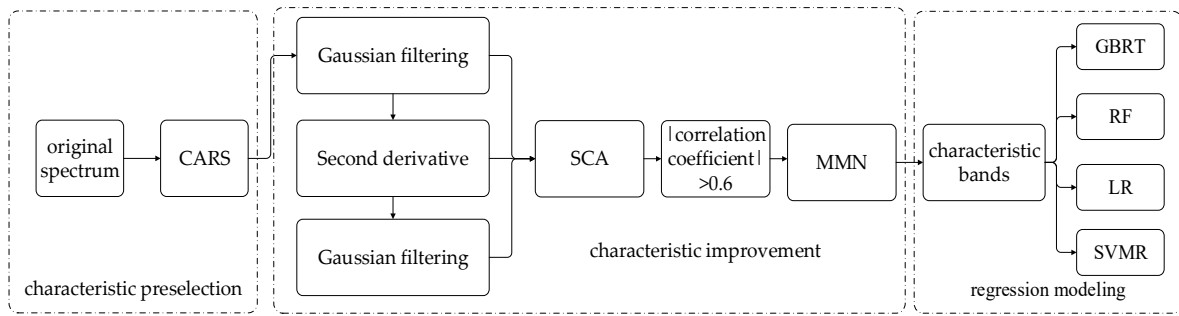

**Figure 2.** Characteristic band selection algorithm flowchart.

### 2.2.2. Gradient Boosting Regression Tree

The gradient boosting algorithm is an optimization algorithm based on error function. Gradient boosting is a machine learning technology that is used to solve classification and regression problems. It generates a strong prediction model by integrating weak prediction models, such as the decision tree. The GBRT algorithm was subsequently developed by Friedman [16]. The core idea is that each calculation is done by a basic model, and the next calculation is undertaken to reduce the residual of the last model and to create a new basic model in the direction of the gradient with reduced residuals [17].

Therefore, by constantly adjusting and optimizing the weight of the weak learner to make it a strong learner, the loss function can be minimized and optimized. The algorithm parameters in this study were set as follows: loss = 'least squres', learning_rate = 0.1, n_estimators = 100, subsample = 1.0, min_samples_split = 2, min_samples_leaf = 1, max_depth = 3, alpha = 0.9, and max_leaf_nodes = 0. Given a set of data points $(x_i, y_i), i = 1, ..., N$, the loss function can be expressed as $g_m(x)$. The input space is split into disjoint regions $R_{1m}, R_{2m}, ..., R_{jm}$, and a constant value is estimated for each region $b_{jm}$. The number of leaf nodes per regression tree is j. The GBRT model and regression tree $g_m(x)$ are expressed as follows:

$$g_m(x) = \sum_{j=1}^{j} (b_{jm}I), x \in R_{jm} \tag{10}$$

$$I(x \in R_{jm}) = \begin{cases} 1, x \in R_{jm}; \\ 0, other; \end{cases} \tag{11}$$

$$L(Y, f(x)) = \sum_{i=1}^{n} (Y - f(x))^2 \tag{12}$$

Step 1: Model initialization:

$$f_0(x) = \underset{\rho}{\text{argmin}} \sum_{i=1}^{n} L(y_i, \rho) \tag{13}$$

Step 2: Iterative generation of M regression trees, where, for m= 1 to M, m represents the mth tree:

(1) For i = 1 to N, i represents the ith sample. The negative gradient value of the loss function is calculated and then used as an estimate of the residual $r_{im}$:

$$r_{im} = -\left[\frac{\partial L(y_i, f_{m-1}(x_i))}{\partial f_{m-1}(x_i)}\right]_{f(x)} = f_{m-1}(x) \tag{14}$$

(2) A regression tree $g_m(x)$ is generated for the residual generated in the previous step. The input space of the m-tree is then divided into J disjoint areas $R_{1m}, R_{2m}, ..., R_{jm}$, and the step size of the gradient drop is calculated:

$$\rho_m = \underset{\rho}{\text{argmin}} \sum_{i=1}^{n} L(y_i, f_{m-1}(x_i) + \rho g_m(x_i)) \tag{15}$$

Step 3: Update of the model, where *lr* represents the learning rate, which is designed to prevent model over-fitting, and to reduce the impact of each base model on the final results.

$$f_m(x) = f_{m-1}(x-1) + lr * \rho_m g_m(x) \tag{16}$$

### 2.2.3. Support Vector Machine Regression

SVM is a nonlinear load forecasting model, the basic principles of which are as follows [18]. Given a set of data points $(x_i, y_i), i = 1, ..., l$, $x_i$ is a factor that is closely related to the forecast, such as the measured spectra of soil from the darkroom. d is the dimension of the selected input variable, $y_i$ is the expected value of the forecast, and $l$ is the total number of known data points. By introducing the Lagrange function, we can express the dual optimization problem as follows:

$$\max\left\{-\frac{1}{2}\sum_{i,j=1}^{l} (\alpha_i - \alpha_i^*)(\alpha_i - \alpha_j^*)k(x_i, y_i) - \varepsilon\sum_{i=1}^{l} (\alpha_i + \alpha_i^*) + \sum_{i=1}^{l} y_i(\alpha_i - \alpha_i^*)\right\} \tag{17}$$

$$st.\sum_{i=1}^{l} (\alpha_i - \alpha_i^*) = 0, \ 0 \le \alpha_i, \ \alpha_i^* \le C, \ i = 1, ..., l \tag{18}$$

where $\alpha_i$ and $\alpha_i^*$ are Lagrange multipliers. Finally, the regression function can be expressed as:

$$f(x) = \sum_{i=1}^{l} (\alpha_i^* - \alpha_i) k(x_i, x) + b \tag{19}$$

### 2.2.4. Random Forest Regression

RF regression is a combined classification model composed of many decision tree regression models, and the parameter set is an independent and identically distributed random vector. Under a given independent variable $X$, each decision tree regression model will have a prediction result [19,20]. Using bootstrap sampling, $k$ samples are extracted from the original measured spectral training set. The sample sizes of these $k$ samples are the same as those of the original training set. $k$ decision tree models are then established for these samples to obtain $k$ regression results. Finally, the $k$ results are averaged to obtain the final prediction.

### 2.3. Accuracy Evaluation

In this paper, the parameters of determination coefficients ($R^2$), root-mean-square error (RMSE), and mean absolute error (MAE) are used to measure the accuracy of the models [21]. The closer $R^2$ is to 1, the better the stability of the model and the higher the degree of fit. RMSE and MAE are used to test the predictive ability of the model. The smaller the RMSE and MAE, the better the predictive ability.

$$R^2 = 1 - \frac{\sum_{i-1}^{n} (\hat{y}_i - y_i)^2}{\sum_{i-1}^{n} (y_i - \overline{y})^2} \tag{20}$$

$$RMSE = \sqrt{\frac{\sum_{i-1}^{n} (y_i - \hat{y}_i)^2}{n}} \tag{21}$$

$$MAE = \frac{1}{m} \sum_{i=1}^{n} |y_i - \hat{y}_i| \tag{22}$$

where n is the number of samples, $y_i$ is the measured value, $\hat{y}_i$ is the predicted value, and $\overline{y}$ is the average of the measured values.

### 2.4. Experimental Procedures

### 2.4.1. Soil Collection and Preparation

The soil sampling points were in the areas of the cities of Daye and Honghu, respectively. Different types of cultivated soils (0–20 cm) were collected in the study areas, including 29 samples in Daye and 63 in Honghu. A chessboard-type sampling method was adopted for the soil collection sites. Foreign bodies such as stones and plant roots in the dried soil were removed, and the soil was then crushed. The crushed soil was then passed through a 2-mm aperture sieve, and then taken out by quartering. The soil was then roller-compacted to pass it through a 0.15-mm aperture sieve [22]. Each soil sample was then divided into two parts for spectral information collection and physical and chemical analysis. The soil samples were digested with nitric acid/hydrochloric acid/perchloric acid and then measured with potassium borohydride/silver nitrate spectrophotometry. Each soil sample was measured three times, and the arithmetic mean was taken as the final As content in the soil. The modal number of the As concentration of the 29 soil samples from Honghu was 34–36 ug/g, which is higher than the modal number value of 8–9 ug/g for the 63 soil samples collected in Daye. The Histograms of the As concentrations are shown in Figure 3.

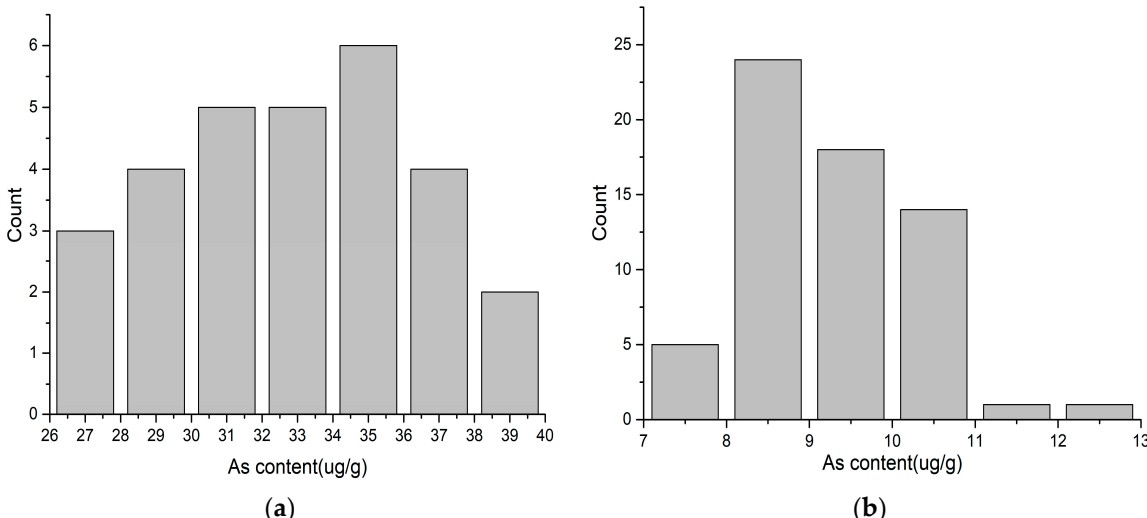

**Figure 3.** Histograms of the As concentrations of the soil samples. (**a**) Soil samples from Honghu. (**b**) Soil samples from Daye.

### 2.4.2. Soil Spectral Reflectance Measurement

In the indoor spectral measurement stage, an SVC HR-1024 field spectrometer was used to measure the spectra of the soil samples from the Honghu area, for which the spectral resolution is as follows: 350 to 1000 nm is 1.5 nm, 1000 to 1900 nm is 3.8 nm, and 1900 to 2500 nm is 2.5 nm. The total number of bands is 990. An ASD FieldSpec 3 field spectrometer was used to measure the spectra of the soil samples from the Daye area, for which the spectral resolution is 1 nm, and the total number of bands is 2151 [23]. The wavelength range of the spectrometers is 350–2500 nm. The soil spectral measurement was carried out in a darkroom. The final processed soil sample was placed on a black petri dish, and the soil surface was flattened to make the surface smooth and easy to measure. The light source was a 1000-watt halogen lamp, with the irradiation direction being 45° from the vertical direction [24]. The light source was set about 30 cm from the surface of the soil sample, with the probe perpendicular to the soil surface and about 10 cm away from the soil sample. White-board calibration was performed on the spectrometers before the measurement. In order to eliminate the instability of the measurement, four spectral curves were measured for each soil sample, after being rotated three times (90° each time) in the same direction, and the actual reflection data were obtained after arithmetic averaging [25].

### 2.4.3. Spectral Pretreatment

Due to the inevitable influence of factors such as the test environment, the instruments, the background of the samples, and stray light in the process of spectrum acquisition, the spectral band edge noise was relatively high [26,27]. In order to reduce the external noise, the noisy edge bands of 350 to 399 nm and 2400 to 2500 nm were removed, and we retained the 400 to 2399nm spectral range for the modeling analysis [28]. In this paper, for the Honghu area, 904 bands were finally retained, and 2000 bands were finally retained for the Daye area. The two sets of soil spectra are shown in Figure 4.

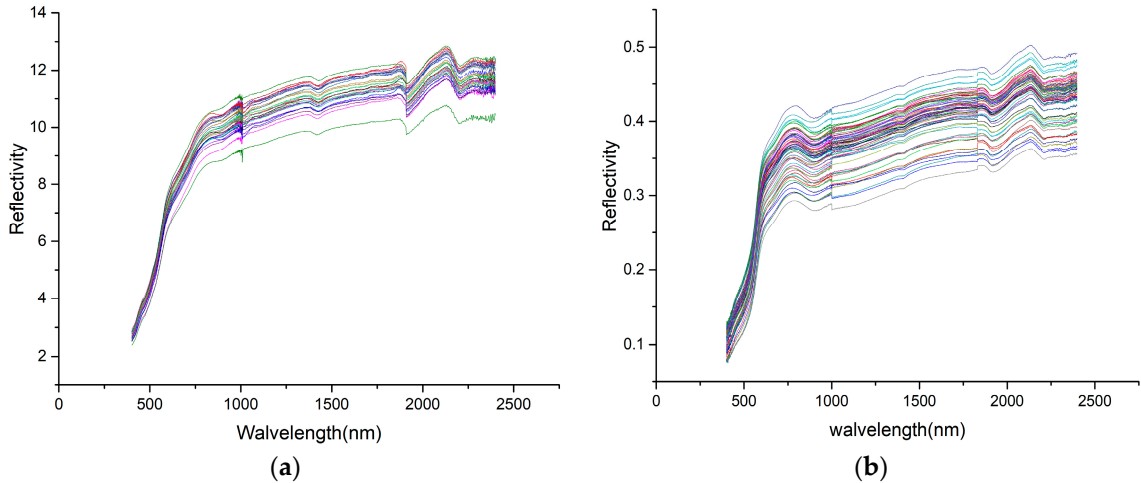

**Figure 4.** The spectra of the soil samples from the different regions. (**a**) Honghu. (**b**) Daye.

### 2.4.4. Calibration Set and Validation Set

Before the modeling, the samples needed to be grouped. One group, called the calibration set, was used for the establishment of the models, and the other group, called the validation set, was used to test the predictive ability of the models. In this study, 92 soil samples were collected from the Honghu and Daye study areas. Nineteen calibration samples and 10 validation samples were selected in the Honghu area, and 41 calibration samples and 22 validation samples were selected in the Daye area. Referring to the *Soil Environmental Quality Standards GB15618-1995* released by the Ministry of Environmental Protection of the People's Republic of China [29], the average value for the Honghu area exceeds the third-class standard (the critical value of soil for the protection of agricultural and forestry production and normal plant growth), and the Honghu area belongs to the polluted area category. The average value for the Daye area is lower than the first-class standard (the limit for soil quality that protects the natural ecology of the area and maintains the natural background), and the Daye area belongs to the unpolluted area category. The difference between the mean, standard deviation, and coefficient of variation of the calibration sets and verification sets is small [30,31]. Therefore, this division can be considered as reasonable and can be used for subsequent modeling. The statistical descriptions of the As concentrations in two study areas are given in Table 1.

**Table 1.** Statistical characteristics of the soil As content.

| Study Area | Sample Type | Sample Size | Minimum (ug/g) | Maximum (ug/g) | Mean (ug/g) | SD | CV (%) | Skewness | Kurtosis |
|---|---|---|---|---|---|---|---|---|---|
| Honghu | Calibration | 19 | 27.33 | 39.21 | 33.44 | 3.34 | 9.98% | 0.03 | −0.49 |
| | Validation | 10 | 26.54 | 37.09 | 31.7 | 3.72 | 11.7% | 0.05 | −1.17 |
| Daye | Calibration | 41 | 7.19 | 12.84 | 9.37 | 1.17 | 12.5% | 0.61 | 0.37 |
| | Validation | 22 | 7.04 | 11.24 | 9.12 | 0.99 | 10.8% | 0.31 | 0.13 |

## 3. Results and Discussion

### 3.1. The CARS-SCA Characteristic Band Selection Algorithm

The CARS algorithm was used to perform rough selection of the original soil spectra, and the number of Monte Carlo samples was set at 100. Figures 5 and 6 show the selection process of spectral variables for the CARS method. In Figures 5a and 6a, the number of variables gradually decreases, and the downward trend is slower and slower; the RMSECV value in Figures 5b and 6b indicates the prediction effect of the PLS model based on the characteristic wavelength of the adaptive reweighted sampling selection; In Figures 5c and 6c, each line represents the change trend of the coefficient of reversion of each wavelength, and the * marks the position with the smallest RMSECV value. After the

*, the RMSECV value starts to increase because the valid variable is deleted. For the Daye area, the trend of the RMSECV obtained by 10-fold cross-validation (CV) is shown in Figure 5b, where it can be seen that the RMSECV value changes significantly. When the number of samples is 62, the RMSECV value is the smallest, which indicates that the model has the highest accuracy. As the sampling runs increase, the RMSECV increases and the model effect deteriorates. Therefore, when the number of samples is 62, the selected spectral band subset is the optimal one, and the RMSECV value is the smallest. In total, 28 characteristic bands (401 nm, 402 nm, 420 nm, 566 nm, 577 nm, 676 nm, 677 nm, 688 nm, 689 nm, 690 nm, 697 nm, 705 nm, 706 nm, 707 nm, 822 nm, 826 nm, 827 nm, 871 nm, 957 nm, 984 nm, 1826 nm, 1906 nm, 2224 nm, 2232 nm, 2344 nm, 2382 nm, 2393 nm, 2394 nm) were obtained. For the Honghu area, the variation trend of the RMSECV obtained by 10-fold CV is shown in Figure 6b. When the number of sampling runs is 60, the RMSECV value is the smallest. This shows that the model has the highest accuracy at this point, and as the sampling runs increase, the increase of the RMSECV indicates that some important variables related to the prediction of As in soil have been eliminated. Therefore, when the sampling frequency is 60 times, the selected spectral band subset is the optimal one, and the RMSECV value is the smallest. In total, 23 characteristic bands (401.9 nm, 405 nm, 408.1 nm, 437.3 nm, 813.6 nm, 817.3 nm, 818.6 nm, 822.3 nm, 823.5 nm, 827.2 nm, 978.9 nm, 997.6 nm, 998.5 nm, 999.4 nm, 1000.3 nm, 1896.8 nm, 1929.2 nm, 1942.9 nm, 2303 nm, 2357.7 nm, 2362.4 nm, 2376.4 nm, 2385.7 nm) were obtained.

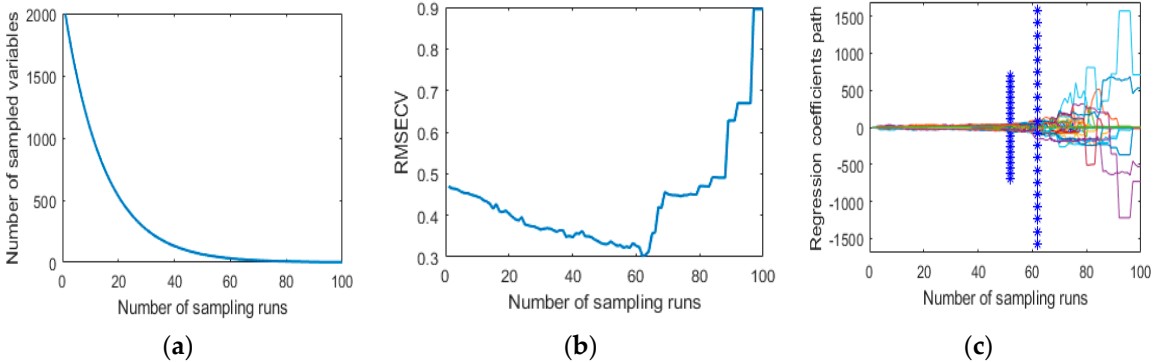

(a)   (b)   (c)

**Figure 5.** (**a**) Change in the number of selected variables with the change in the number of samples runs for the CARS algorithm in Daye. (**b**) Change in the 10-fold root-mean-square error of cross-validation value with the change in the number of sample runs for the CARS algorithm in Daye. (**c**) Change in the regression coefficient variable with the change in the number of samples runs for the CARS algorithm in Daye.

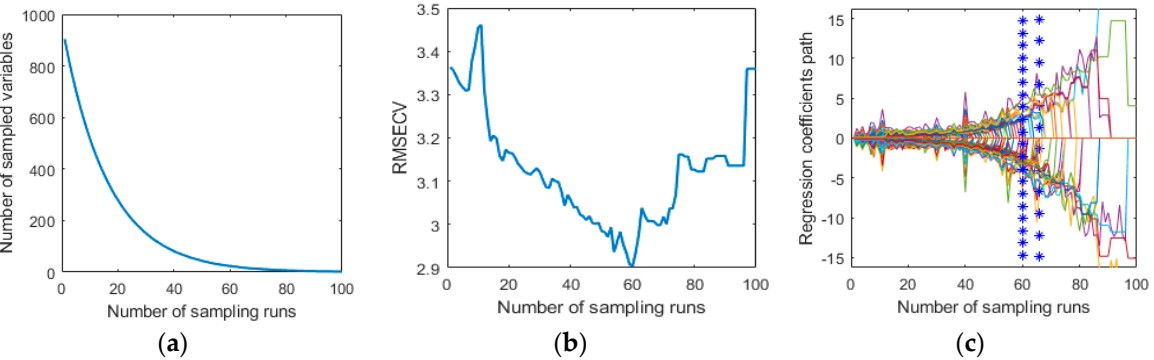

(a)   (b)   (c)

**Figure 6.** (**a**) Change in the number of selected variables with the change in the number of samples runs for the CARS algorithm in Honghu. (**b**) Change in the 10-fold root-mean-square error of cross-validation value with the change in the number of sample runs for the CARS algorithm in Honghu. (**c**) Change in the regression coefficient variable with the change in the number of samples runs for the CARS algorithm in Honghu.

### 3.2. Characteristic Band Improvement

Because the CARS algorithm uses the PLS method as the fitness function, this linear model faces the problem of high dimensional data, which causes the problem of low accuracy. The CARS algorithm does not change the original data, and it is often impossible to use in the case of low-correlation data. It is therefore necessary to improve the characteristic bands. Through the SD, GF and GF again pretreatment methods, the characteristic bands can be effectively enhanced. By extracting Spearman's rank correlation coefficients with absolute values of greater than 0.6 for the subsequent model building, the first *k* correlation-optimized bands were obtained as the characteristic band subset. It can be seen from Figure 7 that the spectral correlation is still low after the CARS rough selection, but the correlation coefficient is greatly improved by the pretreatment. Five enhanced characteristic bands (827.2 nm, 1000.3 nm, 1929.2 nm, 1949.2 nm, 2357.7 nm) were obtained for the Honghu area, and nine enhanced characteristic bands were obtained for the Daye area (402 nm, 420 nm, 566 nm, 577 nm, 689 nm, 690 nm, 697 nm, 2382 nm, 2393 nm), for use in the final modeling.

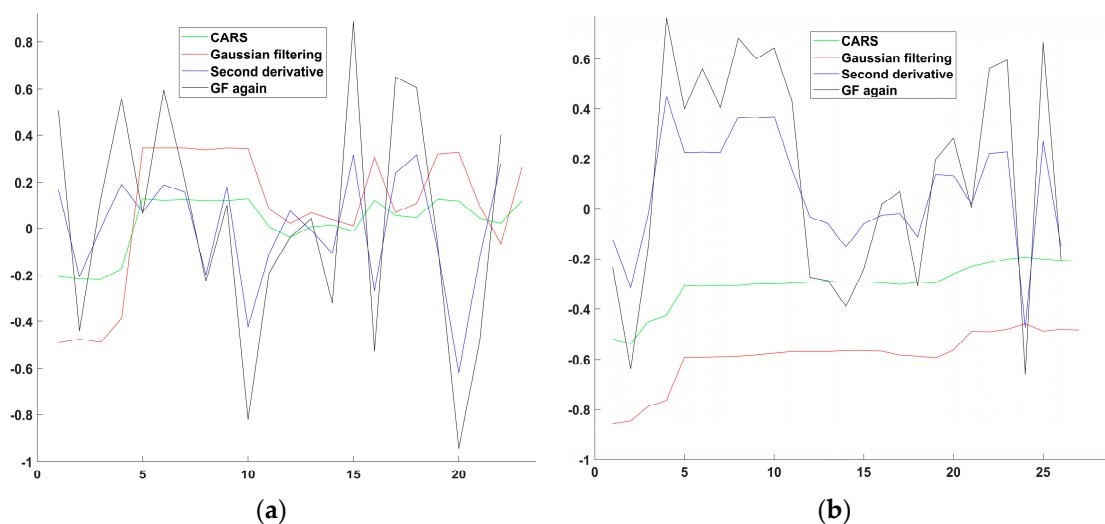

**Figure 7.** Variation of the correlation coefficient after the different pretreatments. (**a**) Honghu; (**b**) Daye.

### 3.3. Comparative Analysis

3.3.1. Analysis of the Results of the Feature Selection Algorithm Based on CARS

In the study, the four different regression models were used to establish the prediction model of soil As content. The closer $R^2$ is to 1, the higher the degree of fitting. The predictive power of the model is measured by RMSE and MAE. The smaller the value, the more accurate the prediction result of the model and the higher the accuracy. The prediction accuracies for the As concentration in the soil obtained with CARS using the four different regression models are given in Table 2. For the Honghu area, the four inversion methods obtain a lower accuracy. For the Daye area, the $R_p^2$ values of the models constructed by the four regression methods are all below 0.5. The reason for this is that the full spectrum contains a lot of redundant and irrelevant information. CARS can remove the redundancy between the characteristic bands, but it cannot improve the correlation between spectral reflectivity and soil heavy metal As, resulting in a lower model accuracy.

**Table 2.** Accuracy validation of the different models.

| Study Area | Characteristic Band Wavelengths (nm) | Modeling Method | $R_p^2$ | $RMSE_p$ | $MAE_p$ |
|---|---|---|---|---|---|
| Honghu | 401.9, 405, 408.1, 437.3, 813.6, 817.3, 818.6, 822.3, 823.5, 827.2, 978.9,997.6,998.5, 999.4, 1000.3, 1896.8, 1929.2, 1942.9, 2303, 2357.7, 2362.4, 2376.4, 2385.7 | GBRT | −0.9364 | 4.9147 | 3.6577 |
|  |  | RF | −0.7230 | 4.6360 | 3.7935 |
|  |  | LR | −0.3553 | 4.1116 | 3.2008 |
|  |  | SVMR | −0.5136 | 4.3452 | 3.5838 |
| Daye | 401, 402, 420, 566, 577, 676, 677, 688, 689, 690, 697, 705, 706, 707, 822, 826, 827, 871, 957, 984, 1826, 1906, 2224, 2232, 2344, 2382, 2393, 2394 | GBRT | 0.2140 | 0.8555 | 0.7089 |
|  |  | RF | 0.4131 | 0.7392 | 0.6488 |
|  |  | LR | 0.3028 | 0.8057 | 0.6568 |
|  |  | SVMR | 0.0046 | 0.9628 | 0.7804 |

### 3.3.2. Analysis of the Results of the Feature Selection Algorithm Based on CARS-SCA

The CARS-SCA algorithm combines the global search capability of CARS with the inversion capability of PLS, it reuses the feature enhancement methods to improve the correlation, and finally uses Spearman's correlation analysis to identify the optimal subsets. It can be seen from Table 3 that good inversion results are obtained with the GBRT regression method. For the Honghu area, the LR prediction effect is poor, and the $R^2$, RMSE, and MAE of the validation set are 0.8914, 1.1638, and 0.9212, respectively. The $R^2$, RMSE, and MAE of the GBRT validation set are 0.9711, 0.6007, and 0.4990, respectively, which are the best prediction results for the study area. For the Daye area, the SVMR prediction effect is poor, and the $R^2$, RMSE, and MAE of the validation set are 0.8691, 0.3491, and 0.1935, respectively. Meanwhile, the $R^2$, RMSE, and MAE of the GBRT validation set are 0.9796, 0.1379, and 0.1078, respectively, which represent the best prediction results for this region. In summary, the $R^2$ values of the validation sets of all the models are all above 0.85, which shows that the overall accuracy is high enough to meet the actual needs, while GBRT is superior to the other three models in stability and predictive ability, and the inversion effect is better.

**Table 3.** Accuracy validation of the different models.

| Study Area | Characteristic Band Wavelengths (nm) | Modeling Method | $R_p^2$ | $RMSE_p$ | $MAE_p$ |
|---|---|---|---|---|---|
| Honghu | 827.2, 1000.3, 1929.2, 1949.2, 2357.7 | GBRT | 0.9711 | 0.6007 | 0.4990 |
|  |  | RF | 0.9557 | 0.7432 | 0.6714 |
|  |  | LR | 0.8914 | 1.1638 | 0.9212 |
|  |  | SVMR | 0.9111 | 1.0531 | 0.8139 |
| Daye | 402, 420, 566, 577, 689, 690, 697, 2382, 2393 | GBRT | 0.9796 | 0.1379 | 0.1078 |
|  |  | RF | 0.9542 | 0.2035 | 0.1562 |
|  |  | LR | 0.9018 | 0.3023 | 0.2269 |
|  |  | SVMR | 0.8691 | 0.3491 | 0.1935 |

### 3.4. Regression Model

In this study, GBRT, RF, LR, and SVMR models were used for the regression. Based on the CARS-SCA characteristic band selection algorithm, the characteristic bands with correlation values greater than 0.6 were used as the independent variables, and the soil As content was used as the dependent variable to construct the soil As model. It can be seen from Figures 8 and 9 that the eight models all obtain a good accuracy. The closer the scatter plot of the measured value and predicted value to the 1:1 line, the higher the inversion accuracy. Most of the samples are closely distributed around the 1:1 line, indicating that the inversion models are accurate. Among them, the GBRT regression model shows the smallest deviation from the 1:1 line, and the degree of fitting is the highest.

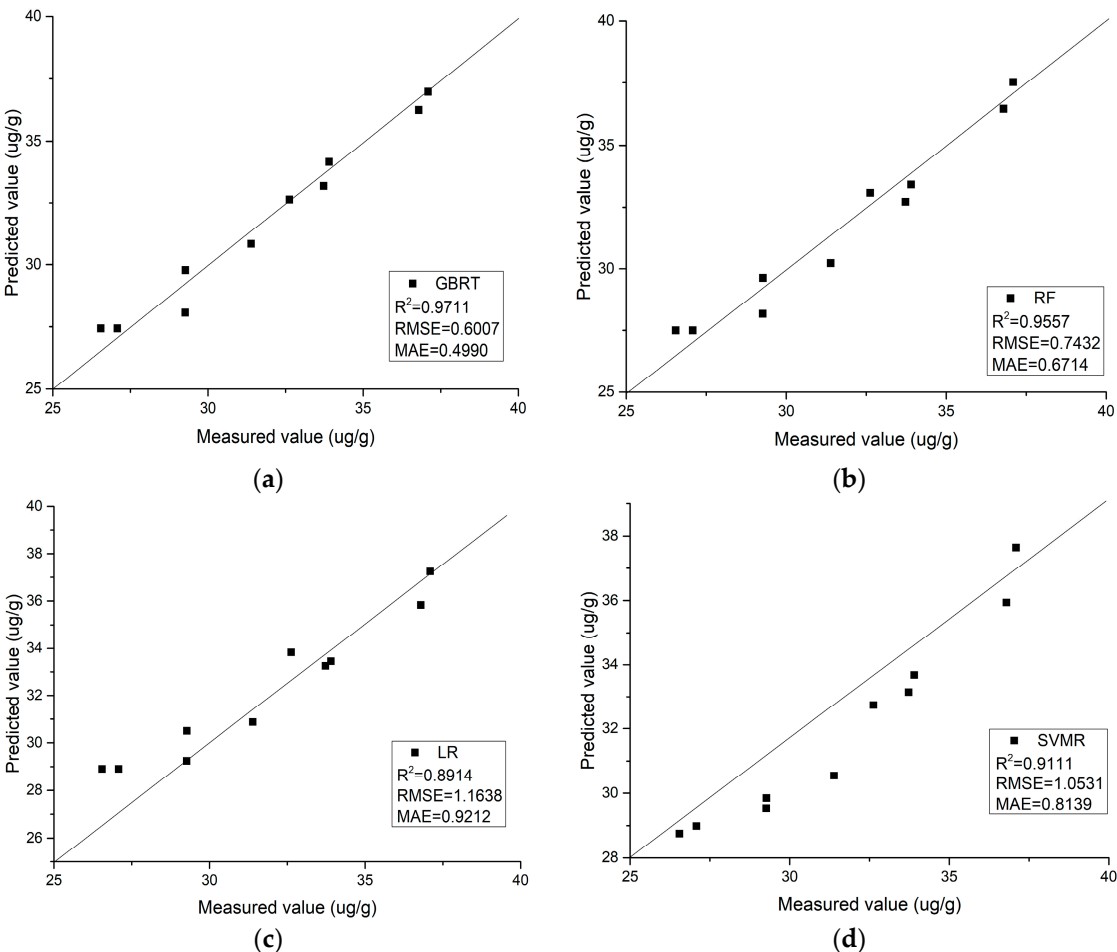

**Figure 8.** A comparison between the measured values and predicted values of the different regression models for Honghu. (**a**) Gradient boosting regression tree model. (**b**) Random forest regression model. (**c**) Linear regression model. (**d**) Support vector machine regression model.

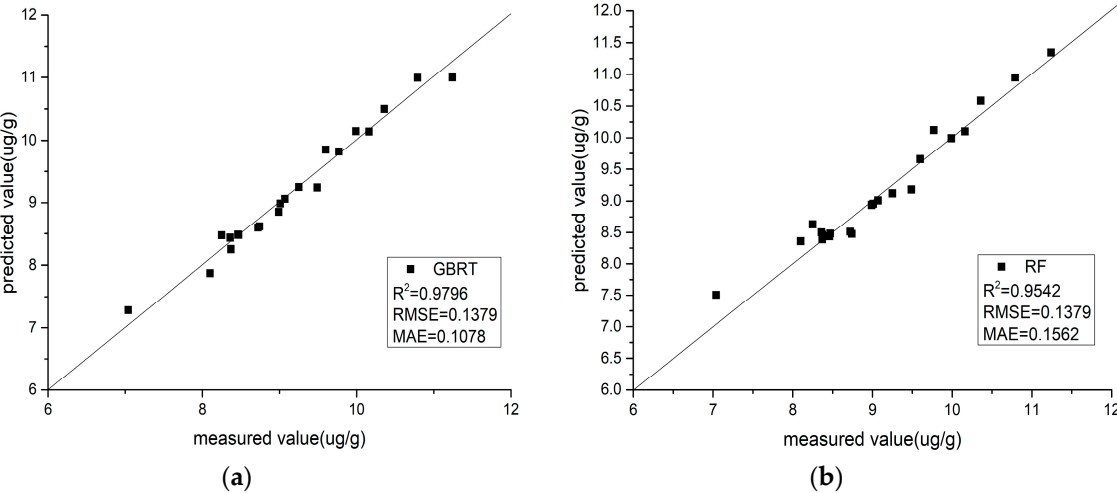

**Figure 9.** *Cont.*

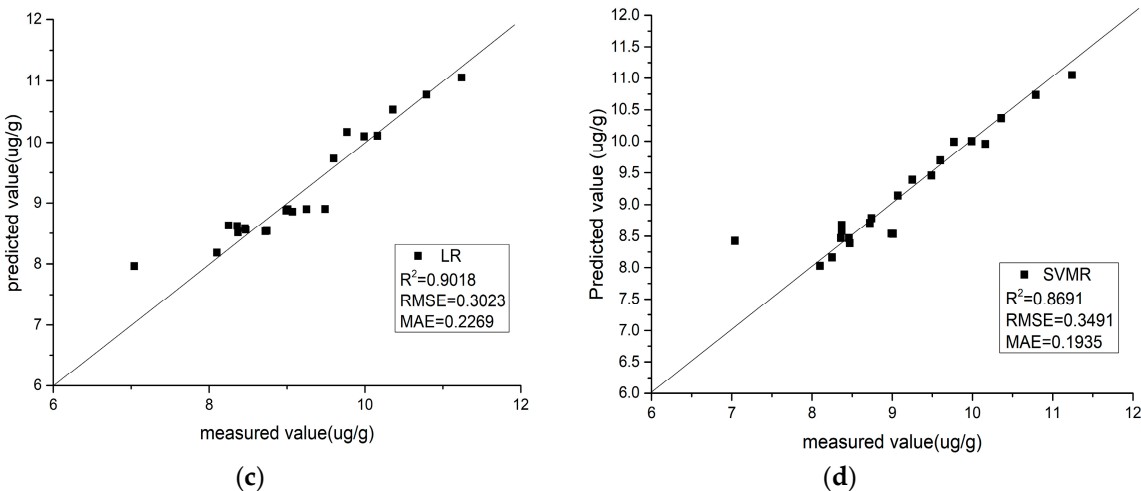

**Figure 9.** A comparison between the measured values and predicted values of the different regression models for Daye. (**a**) Gradient boosting regression tree model. (**b**) Random forest regression model. (**c**) Linear regression model. (**d**) Support vector machine regression model.

## 4. Conclusions

In this study, we took the contaminated soils in the areas of the cities of Daye and Honghu in Jianghan Plain, China, as the research objects. Based on four different model construction methods, we focused on the use of soil reflectance spectral curves to invert the soil As content. The main conclusions are as follows:

(1) The data quality has a significant impact on the modeling effects. By comparing the different regression models based on the CARS and CARS-SCA feature selection algorithms, it was found that CARS coupled with the SCA characteristic band selection algorithm can effectively eliminate irrelevant and redundant information, and can greatly improve the correlation between the spectra and soil heavy metal As content. It was also found that the correlation is greatly improved after the pretreatment, which overcomes the shortcomings of the traditional feature pre-selection algorithm, without changing the original data.

(2) By comparing the four regression methods in the two research areas, it was found that the GBRT model shows a high inversion accuracy and a good generalization ability, indicating that the model has high stability and obtains a high prediction accuracy, which can satisfy the actual forecasting requirements.

(3) The experiments showed that it is feasible to use the CARS-SCA-GBRT algorithm for the spectral analysis of soil heavy metal As content. By analyzing the measured spectra of soil, the problem of information redundancy and poor prediction accuracy in the field of spectral inversion is solved, which will provide a basis for large-scale inversion of soil heavy metal content in the future. In addition, the regional construction of hyperspectral inversion models will also provide a reference for unmanned aerial vehicle, aerospace, and aerial remote sensing.

**Author Contributions:** L.W. and Z.Y. were responsible for the overall design of the study and contributed to the proofreading of the manuscript. Z.Y. performed the experiments, analyzed and interpreted the data, wrote the manuscript, and helped with the proofreading of the manuscript. Y.Z. and X.H. contributed to designing the study and the proofreading of the manuscript. L.Y. contributed to the chemical analysis. Y.Z. analyzed and interpreted the data. All authors read and approved the final manuscript.

**Funding:** This research was funded by the National Key Research and Development Program of China (2017YFB0504202), the National Natural Science Foundation of China (41622107), the Special Projects for Technological Innovation in Hubei (2018ABA078), the Open Fund of the Key Laboratory of Ministry of Education for Spatial Data Mining and Information Sharing (2018LSDMIS05), and the Open Fund of the Key Laboratory of Agricultural Remote Sensing of the Ministry of Agriculture (20170007).

**Acknowledgments:** We gratefully acknowledge the help of the Data Extraction and Remote Sensing Analysis Group of Wuhan University (RSIDEA) in collecting the data. The Remote Sensing Monitoring and Evaluation of Ecological Intelligence Group (RSMEEI) helped to process the data.

**Conflicts of Interest:** The authors declare no conflicts of interest.

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
