# Peer review of "An Improved Gradient Boosting Regression Tree Estimation Model for Soil Heavy Metal (Arsenic) Pollution Monitoring Using Hyperspectral Remote Sensing"

_applsci, doi:10.3390/app9091943_

Round 1
Reviewer 1 Report
This paper proposes a systematic method that can estimate Arsenic (As) pollution degree in the hyperspectral signals. It includes several parts: characteristic band pre-selection, characteristic improvement, and characteristic band selection, and regression modeling. The experiments conducted on field collected hyperspectral soils samples with the chemical ground truth give promising results.
The introduction is well written, and the signal processing pipeline sounds correct. Experimental quantitative results are well presented. However, there are a few issues that are encouraged to fix. My comments are as follows:
Major comments:
1. Because band selection (BS) plays an important role in the proposed method, the authors should clearly explain how the characteristic band selection works. In section 2.2.1 (the introduction of CARS-SCA) Step1, the authors did not provide enough mathematical definitions. For example, the definition of M matrix is not well-defined. What do these Y_{i} vectors in M stand for? And what are the number of spectral samples and the spectral dimensionality? Moreover, in Eq.(1), X vector, \{rho}, \{gamma}, and d are not defined. There are two different M shown in Eq.(1). The author needs to explain all the notations clearly in the revised manuscript, and avoid using duplicate symbols.
2. Step 2 is responsible for characteristic improvement. But the authors did not explain how those methods (i.e., SD, GS, and SNV) can improve the characteristics. What is the output of Step 2? Is there any mathematical notation presenting it (correlation coefficient vector)? Similarly, the operation of Step 3 is unclear, too. In conclusion, the rephrase of Section 2.2.1 is required.
3. In Abstract and Introduction, the authors say that hyperspectral remote sensing has the characteristic of “low correlation”. The reviewer could not understand the meaning. The authors should explain this property and why this property would harm the stability of the inversion model, and how does it affect BS result.
4. In section 3.1, to perform rough spectra selection, how does the sampling works?
5. In section 2.4.2, what are the total number of bands of the raw hyperspectral signals obtained by SVC HR-1024 and ASD FieldSpec 3? Or what is the spectral resolution of these two devices?
6. The experiments of Section 3.3 and 3.4 are performed based on using 5 enhanced bands for Daye data, and 9 enhanced bands for Honghu data. In order to verify the effectiveness of the characteristic band improvement, the authors are encouraged to put the quantitative results of using the rough band sets, that is, 24 bands for Daye data and 28 bands for Honghu data, for the comparison.
Author Response
Please see attachment. Thank you again for your comments and suggestions.

Reviewer 2 Report
The paper presents a method to select features of hyperspectral data for the evaluation of the soil contamination by Arsenic. A combination of methods retrieved from the literature is used to extract robust features, which are used to evaluate the concentration of the pollutant. Two regions are used as examples to vallidate the method. Secondly, the performance different regression techniques are compared.
Remarks:
the aim of the study is not to provide new methods, but to assess the best combination of tools for this specific application. In this sense, providing experimental data represents a strong point of the paper, but on the other hand, being all the methods standard, the general value of the results is not clear.
furthermore, in in spite of the title, the research is at a very early stage, far from being a "remote sensing" method: the prediction of the concentration of As has to be made in laboratory, and the procedure is not simple.
Minor points:
-in the formalization at page 4 (eq. 1) the simbols are not univoque, and n does not correspond to any value. please check
-line 165: please indicate the loss function formula
-lines 179-181: it is not clear to which factor you are referring
-formula (6): please express more clearly the formalization of the problem. please also indicate the meaning of both objective and contraints
-Figure (3): please clarify better in the text the meaning of the figure. for example, what you are accounting for
-line 260: The choice of the the Monte Carlo method for comparing with the performance of the proposed method should be justified. Having to select among 100 frequencies, any random-based choice will be outperformed by a method guided by a correlation analysis
Author Response

(The authors gave the same response as above.)

Round 2
Reviewer 1 Report
Thanks for providing the rebuttal. I would like to recommend accept for this paper.
Author Response
Thanks a lot for your kind comments.The writing of the manuscript was improved by a native speaker and the manuscript has been carefully revised considering about the language and grammar problems.